# Spirituality as a protective factor for chronic and acute anxiety in Brazilian healthcare workers during the COVID-19 outbreak

Julio Cesar Tolentino[1,2]*, Ana Lucia Taboada Gjorup[1,2], Carolina Ribeiro Mello[2,3], Simone Gonçalves de Assis[2,4], André Casarsa Marques[2], Áureo do Carmo Filho[2], Hellen Rose Maia Salazar[2], Eelco van Duinkerken[2,5,6], Sergio Luis Schmidt[2]

**1** Department of Internal Medicine, University Hospital Gaffrée and Guinle, Federal University of the State of Rio de Janeiro, Rio de Janeiro, RJ, Brazil, **2** Department of Neurology, Neurology Post-Graduate Program, University Hospital Gaffrée and Guinle, Federal University of the State of Rio de Janeiro, Rio de Janeiro, RJ, Brazil, **3** Department of Anesthesiology, University Hospital Gaffrée and Guinle, Federal University of the State of Rio de Janeiro, Rio de Janeiro, RJ, Brazil, **4** Department of Studies on Violence and Health, National School of Public Health, Oswaldo Cruz Foundation, Rio de Janeiro, Brazil, **5** Department of Medical Psychology, Amsterdam University Medical Centers, Vrije Universiteit, Amsterdam, The Netherlands, **6** Amsterdam Diabetes Center / Department of Internal Medicine, Amsterdam University Medical Centers, Vrije Universiteit, Amsterdam, The Netherlands

☯ These authors contributed equally to this work.
* julio.tolentino@unirio.br

**Data Availability Statement:** All relevant data are within the manuscript and its Supporting Information files.

## Abstract

### Background

Anxiety symptoms (AS) are exacerbated in healthcare workers (HCWs) during the COVID-19 pandemic. Spirituality is known to protect against AS in the general population and it is a construct that differs from religion. It can be assessed using structured questionnaires. A validated questionnaire disclosed three spirituality dimensions: peace, meaning, and faith. In HCWs we investigated the predictors of chronic anxiety (pre-COVID-19 and during the pandemic) and acute anxiety (only during the pandemic), including spirituality in the model. Then, we verified which spirituality dimensions predicted chronic and acute anxiety. Lastly, we studied group differences between the mean scores of these spirituality dimensions.

### Material and methods

The study was carried out in a Brazilian Hospital. HCWs (n = 118) were assessed for spirituality at a single time-point. They were also asked about AS that had started pre-COVID-19 and persisted during the pandemic (chronic anxiety), and AS that had started only during the pandemic (acute anxiety). The subjects without chronic anxiety were subdivided into two other groups: acute anxiety and without chronic and acute anxiety. Forward stepwise logistic regressions were used to find the significant AS predictors. First, the model considered sex, age, religious affiliation, and spirituality. Then, the analysis were performed considering only the three spirituality dimensions. Group means differences in the spirituality dimensions were compared using univariate ANCOVAS followed by T-tests.

**Funding:** The authors received no specific funding for this work.

**Competing interests:** The authors have declared that no competing interests exist.

## Results

Spirituality was the most realible predictor of chronic (OR = 0.818; 95%CI:0.752–0.890; p<0.001) and acute anxiety (OR = 0.727; 95%CI:0.601–0.881; p = 0.001). Peace alone predicted chronic anxiety (OR = 0.619; 95%CI:0.516–0.744; p<0.001) while for acute anxiety both peace (OR:0.517; 95%CI:0.340–0.787; p = 0.002), and faith (OR:0.674; 95%CI:0.509–0.892; p = 0.006) significantly contributed to the model. Faith was significantly higher in subjects without AS.

## Conclusion

Higher spirituality protected against chronic and acute anxiety. Faith and peace spirituality dimensions conferred protection against acute anxiety during the pandemic.

## Introduction

Anxiety symptoms (AS) are frequently associated with physical and mental illnesses [1–3]. Age and sex are reliable predictors of AS [4–9]. Furthermore, some studies have reported a protective effect of spirituality on AS in the general population [10–16]. Spirituality can involve cognitive and emotional states such as beliefs, motivations, and a sense of gratitude [17–19]. In particular, positive cognitive or emotional aspects of spirituality have been found to be associated with less anxiety [18]. Moreover, González-Sanguino et al., using regression methods, have demonstrated the importance of spirituality as the main protector against the appearance of AS [15].

Previous investigations have shown that AS are exceptionally high in healthcare workers (HCWs) [20]. Given the potential protection of greater spirituality against anxiety [12–16] and the high prevalence of AS among HCWs [20–22], it would be interesting to investigate spirituality as a protector of AS in this specific population. However, the effect of spirituality on AS has not been systematically investigated in HCWs.

During the COVID-19 outbreak, AS have been found to be exacerbated among HCWs worldwide [21–26]. In Brazil, the current pandemic has massively impacted HCWs with a high prevalence of AS [27–29]. A previous cross-sectional study conducted among Brazilian HCWs during the first six months of 2020 showed that most of them experienced a high level of anxiety [27]. Villela et al. also found that the pandemic had a crucial impact on HCWs' work routines, with a high rate of AS [28]. It is noteworthy that spirituality research in Brazil has great relevance due to its cultural context since spirituality and religion are commonplace within Brazilian culture [30, 31]. Indeed, most of the Brazilian population has a religious affiliation [31]. To our knowledge, the relative importance of the potential spirituality protective effect on AS has not been studied among Brazilian HCWs in the current pandemic.

Spirituality encompasses a broader sense of inner peacefulness or harmony, a search for meaning and purpose of life, and how an individual experiences his or her faith [32–34]. It can be measured using well-validated instruments, such as the Functional Assessment of Chronic Illness Therapy-Spiritual Well-being (FACIT-Sp) [35–37]. The FACIT-Sp measures overall spirituality and three psychologically meaningful dimensions: peace, meaning, and faith [36, 38]. These spirituality domains are considered as separate constructs that make up spiritual well-being [37, 39]. Another highlighted aspect is that this multifactorial construct differs from religion [39, 40].

In times of distress, it could be hypothesized that a higher spirituality could predict lower AS in HWCs who are submitted to stressful work situations. Besides, the protection given by the different spirituality dimensions against AS may depend on the moment these symptoms begin. A feeling of peace reflecting an affective dimension of spirituality [35, 36, 41, 42], and it has been associated with mental health [35, 38, 43]. Then, we hypothesized that peace dimension would be a predictor of AS in HCWs who anxiety started before and persisted during the pandemic (chronic anxiety) and those whose anxiety started only during the pandemic (acute anxiety). Given that personal faith can increase psychological resilience [44, 45], we hypothesized that faith would reach the highest score in those without chronic and acute anxiety. In addition, as faith is associated with better coping in stressful times [46–48], we also hypothesized that higher personal faith could be a predictor of less acute anxiety during the pandemic. Therefore, the potential protective effect of spirituality, and its dimensions on AS should be studied in chronic and acute anxiety. However, the effects of each spirituality dimension on the AS among HCWs have not been studied during the COVID-19 outbreak.

The present study aimed to investigate chronic and acute anxiety in HCWs, including spirituality and its dimensions. Firstly, we assessed the AS predictors of chronic. Secondly, we studied the AS predictors of acute anxiety during the pandemic. Thirdly, for chronic and acute anxiety, we analyzed which spirituality dimensions would predict AS. Forthly, we investigated the differences among all the groups (chronic anxiety, acute anxiety and without chronic and acute anxiety), considering the mean scores of the spirituality dimensions that were found to be significant predictors of AS according to the third objective.

## Materials and methods

This observational study was carried out in HCWs from May 12th until July 10th 2020 at a reference Tertiary Hospital for COVID-19 in Rio de Janeiro, Brazil. We included the HCWs of both sexes with ages ranging from 20 to 60 years old. Exclusion criteria: previous or current neurological disorders, uncontrolled clinical conditions, and taking antidepressant, anxiolytic, and antipsychotic medications.

The HCWs who were working in the Hospital during the period of this research were invited to participate in the present study. The researcher explained the study aims and the procedure of data collection. Those who verbally consented to take part in the study were given the questionnaire. The subjects were informed that they could withdraw from the study at any time and return the questionnaire. They were left free to ask questions and to obtain explanations. They all signed the written informed consent.

All filled out a face-to-face questionnaire about spirituality and AS. The presence of AS was assessed in an all-or-none fashion based on a questionnaire that included a question if these symptoms had persisted for a minimum of 6 months. As the first case of COVID-19 in Brazil was reported on February 26[th] 2020, it was possible to identify the participants who presented AS that started before and persisted after this date (chronic anxiety). Then, the sample was divided into two groups according to the presence of AS that had started before COVID-19 and persisted during the pandemic (no chronic anxiety vs. chronic anxiety groups)—first objective. In order to assess the AS directly associated with the pandemic period, we subdivided the sample without anxiety before the pandemic (no chronic anxiety group) into two other groups: subjects that remained without AS all the time (without chronic and acute anxiety group) and those who started AS only during the pandemic (acute anxiety group)—second objective.

Spirituality was assessed through the FACIT-Sp [35, 36, 41, 49, 50]. This scale is a widely used instrument in clinical research to measure spirituality and validated for the Brazilian

population [51, 52]. The FACIT-Sp items emphasize a sense of peacefulness, harmony, meaning or purpose in life, strength and comfort from one's faith or spiritual beliefs´[35, 36, 38]. It is a self-administered questionnaire composed of 12 items, divided equally between three dimensions: peace (items 1, 4, 6, and 7), meaning (items 2, 3, 5, and 8), and faith (items 9, 10,11, and 12). The participant was instructed to indicate how true an affirmative had been for them during the past seven days, using a 5-item response format ranging from not at all (0) to very much (4), except for items four and eight, which must be reverse coded. Total scores range from 0 to 48, with higher scores indicating higher spiritual well-being. The questionnaire also provides scores per dimension. Previous studies have demonstrated the validity and reliability of the FACIT-Sp among Brazilians, e.g., Pereira & Santos (2011) [51] and Luchetti et al. (2013) [52]. These studies have shown good psychometric properties for the Portuguese version of FACIT-Sp, such as high reliability and adequate construct validity [51, 52]. In the present study, the internal reliability of the FACIT-Sp was high [Cronbach's alpha ($\alpha$) = 0.823]. Concerning each spirituality dimension, the Cronbach alphas for the peace, meaning, and faith were 0.865, 0.828, and 0.894, respectively.

For the statistical analysis, quantitative variables were reported as absolute and relative frequencies. The normality of variables was confirmed by assessing the histograms, QQ plots, and Kolmogorov–Smirnov test. Normally distributed continuous variables (age and FACIT-Sp total score) are presented as means and standard deviations and were evaluated using independent $t$-tests. The associations are presented as odds ratios (OR) and 95% confidence intervals (CI 95%).

In the whole sample, a forward stepwise binomial logistic regression was used to predict whether participants could be correctly classified according to the presence or absence of AS that started before the COVID-19 and persited during the pandemic (chronic anxiety group)-first objective. In order to investigate the acute anxiety, we selected only those participants without AS before the pandemic. They were divided into those who remained without AS (without chronic and acute anxiety group) and those who started AS during the COVID-19 outbreak (acute anxiety group). In this subsample, we also investigated the reliable predictor (s) of acute anxiety (second objective). We used the forward stepwise regression method to find the predictors that could contribute significantly to the model. Our model considered sex, age, religious affiliation, and the FACIT-Sp total score for both analyses. We intended to investigate whether these variables could significantly predict less AS among HCWs. Nagelkerke $R^2$ was calculated to estimate the explained variance of the dependent variable, and the Wald test was used to determine the statistical significance of each predictor. The stepwise forward selection included only predictors significant to the model at a probability of F ($p < 0.05$). For both the entire sample and the subsample, forward stepwise logistic regressions were used to determine which spirituality dimensions would predict AS (third objective).

To assess the effect of group (chronic anxiety, acute anxiety, and without chronic and acute anxiety groups) on the spirituality dimensions that were found to predict the presence of AS (forth objective), a MANCOVA was performed to examine group differences on those spirituality dimensions, using age, sex, and religious affiliation as covariates. In the case of a significant overall MANCOVA, post-hoc ANCOVAs for each dependent variable (spirituality dimensions) were checked for statistical significance. In the case of significant ANCOVAs, post-hoc T-tests assessed the presence of significant group differences for each spirituality dimension. Multiple comparisons were correct by the Bonferroni method. For the MANCOVA and each of the ANCOVAs, the η2 (Eta-squared) was computed to calculate the effect size of the results. Cohen has suggested that η2 = 0.01 should be considered a small effect size, 0.06 a medium effect size, and 0.14 a large effect size [53].

For all the tests, significance was set at a p-value < 0.05. SPSS Statistics for Windows, version 25.0 (IBM Corp, 2017), was used for statistical analyses.

The local Research and Ethics committee approved this study (number: 39365120.8.0000.5258) in accordance with the Declaration of Helsinki. Participation in this study was voluntary without monetary or non-monetary compensation. HCWs were invited to take part in this study voluntarily. All of them were informed about the study aims and the entire procedure of data collection. Subjects were assured that all data collected during the research process would be treated confidentially. The volunteers provided written informed consent. They could withdraw from the study at any time by declining to fill in the questionnaire. During the data collection, a researcher was available if any doubts or questions emerged.

## Results

### Predictors of chronic anxiety (first objective)

One hundred and forty-seven were invited, but eight did not agree to participate in this study. Therefore, the initial sample has consisted of 139 subjects. From this sample, 21 were excluded because they reported the use of psychotropic medications. Then, the whole final study sample consisted of 118 subjects. The age ranged from 22 to 60 years (41.9±10.3), and the majority was female (n = 79; 66.9%). Most participants (64.4%) had a religious affiliation. Seventy-two participants (61%) reported chronic anxiety. The FACIT-Sp mean score was statistically higher for the no chronic anxiety group (38.6) as compared to the chronic anxiety group (31.6) [t (116) = 6.0; p<0.001]- S1A Fig and Table 1.

The model included sex, age, religious affiliation, and the FACIT-Sp total score as predictors of chronic anxiety. Spirituality (FACIT-Sp total score) was the only significant predictor in the forward stepwise regression (OR = 0.818; 95% CI: 0.752–0.890; p<0.001). This model was statistically significant [$\chi2(1)$ = 33.2; p<0.001], explaining 33.2% (Nagelkerke R2) of the variance and classifying correctly 71.2% of the cases.

**Table 1. Descriptive statistics of the variables used as predictors for the three groups: Anxiety that started pre-COVID-19 and during the pandemic (chronic anxiety), anxiety that started only during the pandemic (acute anxiety), and without chronic and acute anxiety group.**

| First Objective | No chronic anxiety (n = 46) | Chronic anxiety (n = 72) | p-value |
|---|---|---|---|
| Sex female, n (%) | 27 (58.7%) | 52 (72.2) | 0.2 |
| Age (Years), Mean (SD) | 45.4 (10.4) | 39.7 (9.5) | 0.003 |
| Religious affiliation, n (%) | 29 (63.0) | 47 (65.3) | 0.8 |
| FACIT-Sp (score), Mean (SD) [a] | 38.6 (4.5) | 31.6 (6.9) | <0.001 |
| **Second objective** | **Without chronic and acute anxiety (n = 21)** | **Acute anxiety (n = 25)** | **p-value** |
| Sex female, n (%) | 11 (52.4) | 16 (64.0) | 0.6 |
| Age (Years), Mean (SD) | 46.4 (11.5) | 44.6 (9.5) | 0.6 |
| Religious affiliation, n (%) | 12 (57.1) | 17 (68) | 0.5 |
| FACIT-Sp (score), Mean (SD) [b] | 41.3 (3.9) | 36.2 (3.7) | 0.001 |

Abbreviations: HCWs = healthcare workers; AS = anxiety symptoms; SD = standard deviation; p = proof value; OR = Odds Ratio; CI = confidence interval.

[a] The FACIT-Sp score was the most reliable predictor of chronic anxiety in the forward stepwise regression method (OR = 0.818; 95% CI: 0.752–0.890; p<0.001)

[b] The FACIT-Sp score was the most reliable predictor of acute anxiety during the pandemic in the forward stepwise regression method (OR = 0.727; 95% CI:0.601–0.881; p = 0.001).

### Predictors of acute anxiety (second objective)

To investigate the relationship between spirituality and acute anxiety during the current outbreak, we selected all subjects without AS starting before the pandemic (n = 46). Of these 46 participants, 25 (54.3%) started AS only during the pandemic (acute anxiety group), and 21 continued without AS (without chronic and acute anxiety group). In this subsample, the FACIT-Sp total mean score was statistically higher in participants that remained without AS (41.2) as compared to acute anxiety group (36.2) [t(44) = 4.4; p<0.001]- S1B Fig and Table 1.

The model included sex, age, religious affiliation, and spirituality (FACIT-Sp total score). Spirituality was the only significant predictor of acute anxiety in the forward stepwise regression (OR = 0.727; 95% CI:0.601–0.881; p = 0.001). This model was statistically significant [χ2 (1) = 13.6; p<0.001], explaining 38.5% (Nagelkerke R2) of the variance and classifying correctly 69.6% of the cases.

### Dimensions of spirituality (third objective)

Peace was the only predictor of chronic anxiety (OR = 0.619; 95% CI: 0.516–0.744; p<0.001). This model was statistically significant [χ2(1) = 40.4; p<0.001], explained 39.3% of the variance, and classifying correctly 78.8% of the cases- S2A Fig.

Both peace (OR:0.517; 95% CI: 0.340–0.787; p = 0.002) and faith (OR:0.674; 95%CI:0.509–0.892; p = 0.006) dimensions predicted acute anxiety- S2B Fig. This model was statistically significant [χ2(2) = 23.4; p<0.001], explained 53.3% of the variance, and correctly classified 76.1% of the cases.

### Comparison of spirituality dimensions that were predictors of anxiety (fourth objective)

After adjusting for age, sex, and religious affiliation, the MANCOVA showed a significant effect of group (chronic anxiety, acute anxiety, and without chronic and anxiety group) on peace and faith dimensions (F = 15.3, df = 4/222, p<0.001, η2 = 0.216). The univariate tests showed that group affected peace (F = 29.1, df = 2/112, p<0.001, η2 = 0.342), and faith (F = 5.4, df = 2/112, p<0.006, η2 = 0.88). The t-tests showed significant means differences in all comparisons for the peace dimensions. Regarding the faith dimension, the mean scores were significantly higher in the without chronic and acute anxiety group compared to the chronic anxiety and acute anxiety groups.

## Discussion

We found a high prevalence of AS in HCWs. Our findings are in line with recent studies on the frequency of AS in HCWs [23, 24, 54, 55]. Additionally, we reported a high rate of acute anxiety (54.3%) during the pandemic. These data may reflect the specific psychological impact of COVID-19 on HCWs.

Spirituality was the most reliable predictor of acute and chronic anxiety. We showed that the peace spirituality dimension was a significant predictor of less AS among HCWs irrespective of whether these symptoms were chronic or acute during the pandemic. Moreover, the faith dimension significantly predicted AS that started only during the pandemic. This finding indicates the importance of personal faith as an additional protective factor for acute anxiety in the current outbreak.

### Predictors of chronic anxiety (first objective)

We used the forward stepwise regression method to find the best predictors of the AS. The stepwise forward selection included only predictors that enhanced the significance of the

model at a probability of F (p < 0.05). Based on this approach, sex, age, and religious affiliation were not included as predictors in the logistic regression model (forward regression method). Using this method, the spirituality variable was enough to predict AS adequately. Our results about religion probably reflect that religious affiliation is a construct that is conceptually different from spirituality [39, 40]

The present data revealed that high spirituality protects against chronic anxiety. This is supported by other studies that have shown the association between greater spiritual well-being and lower AS in the general population [15, 16, 56].

The FACIT-Sp mean scores were 38.6 and 31.6 in no chronic and chronic groups, respectively. As a FACIT-Sp score above 36 points has been considered high spiritual well-being [57, 58], we inferred that most subjects in the no chronic group had high spirituality based on this cutoff.

## Predictors of acute anxiety (second objective)

Spirituality was showed to be the only reliable predictor of acute anxiety. As there is an increase in the number of subjects with anxiety in times of distress, it is conceivable that higher spirituality should be needed to overcome the demands of a new stressful situation. The FACIT-Sp mean score in the group without chronic and acute anxiety reached 41.3 points, well above the 36 points (the limit for a high spirituality) [57, 58]. In contrast, the FACIT-Sp score was below this cutoff point in most subjects of the acute anxiety group. As described for chronic anxiety, greater spiritual well-being also protected against acute anxiety during the pandemic.

## The effect of spirituality on chronic and acute anxiety (first and second objectives)

We demonstrate the impact of spiritual well-being among HCWs in chronic and acute anxiety (first and second objectives). High spirituality involves positive emotions and spiritual beliefs, which may provide a better psychological adaptation against AS [18]. Another possible reason is that spirituality has been shown to be a consistent resilience factor [59]. As a higher spirituality was the most reliable predictor of less acute anxiety, we can infer that HCWs with increased spiritual well-being tend to develop internal mechanisms that help them cope with the adversities associated with the current pandemic.

A potential mechanism for spirituality protection on AS could be related to the neurobiological substrates of spirituality [60–63]. Although multiple brain regions may contribute to spirituality, the parietal cortex is arguably the most frequently implicated brain region related to spirituality [63]. Furthermore, the medial parietal region appears functionally connected with lateral areas of the parietal cortex associated with self-reference or a felt sense of connection outside of oneself [64]. Rim et al. (2019) systematically reviewed the relationship between spirituality and electroencephalography, structural and functional neuroimaging [65]. This and others studies have demonstrated several brain regions associated with spiritual well-being, such as parietal cortex, medial frontal cortex, orbitofrontal cortex, precuneus, posterior cingulate cortex, default mode network, and caudate [63, 65–67].

## Effect of the spirituality dimensions on AS (third objective)

The peace dimension of spirituality was the only significant predictor of chronic anxiety. Our results agree with previous studies in different populations, in which the sense of peace was associated with less AS [11, 68, 69]. This finding can be explicated because the peace subscale is an affective expression of spirituality, reflecting a sense of inner harmony [35, 36, 41, 42],

and this dimension has been specifically related to mental health [35, 38, 43]. However, feeling at peace alone did not protect against the acute anxiety during the current outbreak. We found that a high faith score was also necessary to protect against acute anxiety during the pandemic. This result may indicate that when HCWs cope with anxiety triggers during the pandemic, their faith is important to reduce AS.

It should be mention that personal faith is more subjective and deeper than religious affiliation [40, 70]. Our finding may have some possible explanations. First, faith can be a source of hope for the future [45, 47, 71]. Second, faith can provide an optimistic worldview [44, 72], which is inversely associated with AS. Finally, cultivating expressions of faith can increase psychological resilience [45] and, consequently mitigate AS [73].

Taken together, it is possible that a sense of inner peace or harmony and, especially, personal faith have served to better coping with stress in the face of uncertainties associated with the pandemic.

## Comparison of spirituality dimensions that were predictors of anxiety (fourth objective)

As expected, the subjects without chronic and acute anxiety presented the highest mean score for inner peace. The peace dimension was greater in acute anxiety as compared to chronic anxiety. This finding may indicate that mental health was better in the subjects without chronic anxiety.

Regarding the faith dimension, HCWs without anxiety had significantly higher faith scores compared to those with chronic and acute anxiety. These data suggest possible protection of faith against AS in this specific population. In particular, our results highlighted the role faith played in the pandemic. Only the group of HCWs with the highest faith protection could remain without AS during the pandemic. This finding supports the relevance of faith as a predictor for better psychological coping to the current COVID-19 outbreak.

## Limitations, strengths, and future directions

A limitation of the present study is the small number of subjects in the subsample of HCWs. Although the HCWs of the present study can be considered a representative sample of Brazilian HCWs, future studies should be conducted in a larger sample of HCWs recruited from various areas in Brazil. A strength is that the questionnaires were applied face-to-face by the researchers since most previous studies were carried out by e-mail, social media, or online websites.

A practical implication of our findings is that it would be useful to promote enhanced screening for AS and address spirituality in HCWs, particularly during stressful times. Studies of medical and psychiatric patients report that the majority have spiritual issues, and most of those needs currently go unmet [33, 39, 74]. Therefore, health professionals may engage HCWs in a dialogue that allows them to talk about their spirituality. Additionally, based on the present study, it may be useful to take spiritual assessments during the inclusion of HCWs in preventive and therapeutic mental health programs.

From a public health perspective, future research should focus on potential interventions based on spirituality in HCWs. In other populations, psychosocial interventions have proved effective in improving spirituality [75–78], such as spiritually integrated psychotherapy [79, 80], yoga [78], and meditation [78, 79]. In addition, a previous meta-analysis found that faith-adapted cognitive-behavioral therapy (CBT) may outperform both control conditions and standard CBT in the treatment of anxiety [81]. Our data highlight the protective role of faith and inner peacefulness. Therefore, psychosocial interventions in HCWS during the pandemic

should consider these two spirituality dimensions. These approaches may lower the impact of anxiety triggers on HCWs.

## Conclusions

The present study shows that spirituality can be considered a protective factor against AS in HCWs, irrespective of whether the symptoms started before and during the pandemic. This finding suggests that HCWs with high spiritual well-being tend to develop internal mechanisms that help them cope with psychological stress.

Additionally, we found that spirituality dimensions influenced the AS differently according to the starting time of symptoms. Peace spirituality dimension was the best predictor of acute and chronic anxiety, whereas the faith dimension emerged together with peace to confer protection on acute anxiety starting during the COVID-19 outbreak. These data suggest that a personal faith can represent an important coping strategy facing a severe stressful moment among HCWs. Therefore, interventions based on improving spiritual well-being and considering these spirituality dimensions could be designed and implemented in HCWs.

## Supporting information

**S1 Fig. The mean FACIT-Sp total score for each group.** (A) FACIT-Sp total score is statistically significantly higher in the no chronic anxiety group compared to the chronic anxiety group (*p<0.001), indicating a lower spirituality among the subjects experiencing anxiety that started pre-COVID-19 and during the pandemic (B) The boxplot indicates that the FACIT-Sp total score is also statistically significantly higher in the group without chronic and acute anxiety (*p<0.001) as compared to the acute anxiety group.
(TIFF)

**S2 Fig. Effects of spirituality dimensions on anxiety symptoms according to chronic and acute anxiety.** (A) The peace was the only spirituality dimension protective of chronic anxiety (B) Peace and faith spirituality dimensions are protective factors against acute anxiety during the pandemic.
(TIFF)

## Acknowledgments

We would like to thank the study participants.

## Author Contributions

**Conceptualization:** Julio Cesar Tolentino, Ana Lucia Taboada Gjorup, Carolina Ribeiro Mello, Simone Gonçalves de Assis, André Casarsa Marques, Áureo do Carmo Filho, Hellen Rose Maia Salazar, Eelco van Duinkerken, Sergio Luis Schmidt.

**Data curation:** Julio Cesar Tolentino, Ana Lucia Taboada Gjorup, Carolina Ribeiro Mello, Simone Gonçalves de Assis, André Casarsa Marques, Áureo do Carmo Filho, Hellen Rose Maia Salazar, Eelco van Duinkerken, Sergio Luis Schmidt.

**Formal analysis:** Julio Cesar Tolentino, Eelco van Duinkerken, Sergio Luis Schmidt.

**Investigation:** Julio Cesar Tolentino, Ana Lucia Taboada Gjorup, Áureo do Carmo Filho, Hellen Rose Maia Salazar, Eelco van Duinkerken, Sergio Luis Schmidt.

**Methodology:** Julio Cesar Tolentino, Eelco van Duinkerken, Sergio Luis Schmidt.

**Supervision:** Julio Cesar Tolentino, Eelco van Duinkerken, Sergio Luis Schmidt.

**Visualization:** Carolina Ribeiro Mello, André Casarsa Marques, Áureo do Carmo Filho, Hellen Rose Maia Salazar, Eelco van Duinkerken, Sergio Luis Schmidt.

**Writing – original draft:** Julio Cesar Tolentino, Eelco van Duinkerken, Sergio Luis Schmidt.

**Writing – review & editing:** Julio Cesar Tolentino, Ana Lucia Taboada Gjorup, Carolina Ribeiro Mello, Simone Gonçalves de Assis, André Casarsa Marques, Áureo do Carmo Filho, Hellen Rose Maia Salazar, Eelco van Duinkerken, Sergio Luis Schmidt.

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
