## [Decision Letter · Decision Letter 0]

20 Aug 2021

PONE-D-21-18892

Spirituality as a protective factor for anxiety symptoms in healthcare workers during the COVID-19 outbreak

PLOS ONE

Dear Dr.Tolentino, 

Thank you for submitting your manuscript to PLOS ONE. After careful consideration, we feel that it has merit but does not fully meet PLOS ONE’s publication criteria as it currently stands. Therefore, we invite you to submit a revised version of the manuscript that addresses the points raised during the review process.

Please pay particular attention to addressing the extensive methodological issues raised by Reviewer 1.

We look forward to receiving your revised manuscript.

Kind regards,

Rosemary Frey

Academic Editor

PLOS ONE

Journal Requirements: 

2. Peer review at PLOS ONE is not double-blinded (https://journals.plos.org/plosone/s/editorial-and-peer-review-process). For this reason, authors should include in the revised manuscript all the information removed for blind review, including names of institutions.

Reviewers' comments:

Reviewer's Responses to Questions

**Comments to the Author**

1. Is the manuscript technically sound, and do the data support the conclusions?

Reviewer #1: Partly

Reviewer #2: Yes

2. Has the statistical analysis been performed appropriately and rigorously? 

Reviewer #1: N/A

Reviewer #2: Yes

3. Have the authors made all data underlying the findings in their manuscript fully available?

Reviewer #1: No

Reviewer #2: Yes

4. Is the manuscript presented in an intelligible fashion and written in standard English?

Reviewer #1: Yes

Reviewer #2: Yes

5. Review Comments to the Author

Reviewer #1: PONE-D-21-18892

I am pleased to read and review manuscript ID PONE-D-21-18892 entitled " Spirituality as a protective factor for anxiety symptoms in healthcare workers during the COVID-19 outbreak". The study is interesting; however, I consider that specific questions need to be addressed to improve its presentation, which I mention above:

1. The abstract contains sufficient background to understand the problem under investigation. However, please focus and provide the gap statements briefly in your abstract.

2. Line 34, Page 2. Age and sex are not main variable, suggest not include in the abstract.

3. Please provide your result with related the spirituality as a protective factor for anxiety symptoms in result section-abstract.

4. Please consistent to use the word “before and during the pandemic” or “non-pandemic and pandemic periods”

5. Concerning the introduction section, more specific information is needed to provide the novelty, including what and why this study is important among Brazilian healthcare workers (HCWs) with specific spiritual and anxiety symptoms.

6. Please more clearly about the sentences in line 69-70 “However, the effects of each spirituality dimension on the AS of HCWs have not been studied during the COVID-19 outbreak”. Is it valid or true based on the previous study?

7. Line 71-15. The present study aimed to investigate AS predictors in HCWs, including spirituality, considering two different starting times for the AS. First, we assessed AS starting before the COVID-19 outbreak (first objective). Thereafter, we studied the predictors for AS conversion during the pandemic (second objective.) Finally, for each period, we analyzed which spirituality dimensions would predict AS (third objective).

Considering your objective please provide your results based on the objectives in the abstract section, especially the third objective?

8. In line 77-78, Page 3. This study was carried out in HCWs from 12th May until 9th July 2020 at a reference University Hospital for COVID-19 in Rio de Janeiro, Brazil. Based on the time. Please more detail about the time (starting before the COVID-19 outbreak-first objective; and during the pandemic-second objective) or pandemic and pandemic period

9. Line 78-80. Exclusion criteria: age below 20 years or above 60 years, previous or current neurological disorders, uncontrolled clinical conditions, and taking antidepressant, anxiolytic, and antipsychotic medications.

a. Please measure your criteria exclusion age bellow 20 years-related the HCWs or general population?

b. How about the authors clarified or measure the validity of previous or current neurological disorders, uncontrolled clinical conditions, and taking antidepressant, anxiolytic, and antipsychotic medications?

10. Please provide more detail about the validity and reliability FACIT-Sp among Brazilian as well as provide your validity and reliability (based on your results) each domain because the author used each domain

11. Line 95-97. “Normally distributed continuous variables (age and FACIT-Sp total score) are presented as the means and standard deviations and were evaluated using an independent t-tests. As well as “The associations are presented as odds ratios (OR) and 95% confidence intervals (CI 95%)”. Please provide the result, could be easier to use the table related the all objective (before, during the pandemic and each period).

12. The methods: needs to be more complete, I suggest the study design, the specification of the instruments with the references with the best-fitting model.

13. Please provide references related to the minimum sample size and the kind of assessment of sample size method

14. Methods in this manuscript are generally too vague. They need to be more descriptive and clearer in defining inclusion the subjects.

15. Line 111, please provide the detail of ethical approval ethic of this study.

16. In the discussion section, please describe the reason or mechanism of spirituality might protective anxiety symptoms.

17. Please provide data value of sex, age, and religious affiliation based on the objective

18. Line 160. “Sex, age, and religious affiliation were not included in the logistic regression equation because we have applied the forward analysis”. Please provide more detail about the reason and previous studies to similar or contrast about the findings among HCWs (don’t general population).

19. Should include potential other confounding variables that could be related to spiritual and anxiety symptoms. The data did not represent the entire population of health-care workers and the services in which they worked (intensive care, primary care. . .), also, did not include other variables such as whether the participants had had any personal experience of loss or illness due to COVID in their family or friends, and, as a result, the findings cannot be used to make useful generalizations regarding health-care workers as a whole, or to determine specific variables’ correlations with specific groups of health-care workers. A larger sample of health-care workers recruited from various areas in Brazil is needed to verify the results.

Reviewer #2: It would be best if the discussion and conclusion sections could be expanded upon to allow for a more robust discussion of results and their direct implications of future practices for the well being of HCWs. The manuscript was very robust with statistical analysis that showed the implications of spirituality on AS, however lacked a next step for HCW. This would be helpful to add to the manuscript.

6. PLOS authors have the option to publish the peer review history of their article (what does this mean?). If published, this will include your full peer review and any attached files.

Reviewer #1: No

Reviewer #2: No

---

## [Author Response · Author response to Decision Letter 0]

10 Oct 2021

Dear Emily Chenette, PhD - Editor-in-Chief – PLOS ONE

Dear Rosemary Frey, PhD - Academic Editor – PLOS ONE

We are glad to know that, based on the advice received, our manuscript entitled “Spirituality as a protective factor for anxiety symptoms in healthcare workers during the COVID-19 outbreak " has merit for publication. We thank the Editor and two anonymous reviewers for the time spent reviewing our manuscript and for the valuable input given to improve it. 

We send a marked-up copy of the revised version (RV) that highlights changes made to the original version (“Revised Manuscript with Track Changes”). The changes can be found in red letters. The related question raised by the reviewers (reviewer 1- R#1 and reviewer 2- R#2) can be found highlighted in yellow (R#1) and green (R#2) on the right margin of the “Revised Manuscript with Track Changes”. 

Please find below the detailed point-by-point responses to all the interesting and constructive questions (Q) raised by the two reviewers. 

Reviewer #1:

“I am pleased to read and review manuscript ID PONE-D-21-18892 entitled " Spirituality as a protective factor for anxiety symptoms in healthcare workers during the COVID-19 outbreak". The study is interesting; however, I consider that specific questions need to be addressed to improve its presentation, which I mention above:”

We greatly appreciate R#1 for his/her meaningful comments on our manuscript. As R#1 points out, we agree that specific questions need to be addressed to improve our manuscript. So, we have answered the 19 questions raised by R#1 described below.

Q1 R#1: “The abstract contains sufficient background to understand the problem under investigation. However, please focus and provide the gap statements briefly in your abstract.”

Answer Q1 R#1: We are glad that “The abstract contains sufficient background to understand the problem under investigation.”. We agree with the following interesting comment made by R#1:"please focus and provide the gap statements briefly in your abstract.". We have included more focus and a brief gap statement in RV of the abstract. Please see line 30, page 2. 

Q2 R#1: “Line 34, Page 2. Age and sex are not main variable, suggest not include in the abstract.”

Answer Q2 R#1: We entirely agree with the point raised by R#1. We have removed the following sentence in the RV of abstract: "Age and sex are AS predictors in HCWs”.

Q3 R#1: ‘’ Please provide your result with related the spirituality as a protective factor for anxiety symptoms in result section-abstract.”

Answer Q3 R#1: We agree with this interesting comment raised by R#1. Then, we provided our result about spirituality as a protective factor for AS starting before and during the pandemic. Please, see lines 41-43 in the results section of RV abstract.

Q4 R#1: “Please consistent to use the word “before and during the pandemic” or “non-pandemic and pandemic periods”

Answer Q4 R#1: We thoroughly agree with R#1. As the reviewer points out, we replaced “non-pandemic and pandemic periods” with “before and during the pandemic” in the RV.

Q5 R#1: “Concerning the introduction section, more specific information is needed to provide the novelty, including what and why this study is important among Brazilian healthcare workers (HCWs) with specific spiritual and anxiety symptoms”

Answer Q5 R#1:: We thank R#1 for this thoughtful comment. In the RM introduction section, we included why our study is important among Brazilian HCWs, focusing on anxiety symptoms and potential protective factor of spirituality against these symptoms in HCWs. Please see lines 53-55, first paragraph (page 3), and lines 57-62, second paragraph (page 3).

Q6 R#1: “Please more clearly about the sentences in line 69-70 “However, the effects of each spirituality dimension on the AS of HCWs have not been studied during the COVID-19 outbreak”. Is it valid or true based on the previous study?”

Answer Q6 R#1: To our knowledge, the effect of spirituality dimensions (peace, meaning, and faith) on AS in HWCs have not been studied in the current pandemic. Then, it is valid and true based on the literature review of previous studies. Please see lines 71-75 (pages 3 and 4).

Q7R#1: “Line 71-15. The present study aimed to investigate AS predictors in HCWs, including spirituality, considering two different starting times for the AS. First, we assessed AS starting before the COVID-19 outbreak (first objective). Thereafter, we studied the predictors for AS conversion during the pandemic (second objective.) Finally, for each period, we analyzed which spirituality dimensions would predict AS (third objective).

Considering your objective please provide your results based on the objectives in the abstract section, especially the third objective?”

Answer Q7 R#1: We would like to thank R#1 for this interesting suggestion. First, we described our three objectives in the background section of the RV abstract (lines 31-34). After, we provided our results based on each objective in the results section of the revised abstract (lines 41-45).

Q8R#1: “In line 77-78, Page 3. This study was carried out in HCWs from 12th May until 9th July 2020 at a reference University Hospital for COVID-19 in Rio de Janeiro, Brazil. Based on the time. Please more detail about the time (starting before the COVID-19 outbreak-first objective; and during the pandemic-second objective) or pandemic and pandemic period”

Answer Q8 R#1: We thank the reviewer for pointing this out. Based on our study time, all eligible HCWs filled out a face-to-face questionnaire about spirituality and AS. The presence of AS was assessed in an all-or-none fashion based on a questionnaire that included a question if these symptoms had persisted for a minimum of 6 months. As the first case of COVID-19 in Brazil was reported on February 26th 2020, it was possible to identify the participants who presented AS that started before this date. Then, the sample was divided into two groups according to the presence of AS that had started before the pandemic (no-AS and AS groups)- first objective. In order to assess the AS directly associated with the pandemic period, we subdivided the sample into two other groups: no-AS subjects that remained without AS (non-converted group) and subjects who converted to AS (converted group)- second objective. Please see lines 91-97, fifth paragraph (page 4).

Q9 R#1: Line 78-80. Exclusion criteria: age below 20 years or above 60 years, previous or current neurological disorders, uncontrolled clinical conditions, and taking antidepressant, anxiolytic, and antipsychotic medications.

a. Please measure your criteria exclusion age bellow 20 years-related the HCWs or general population?

Answer Q9a R#1: We do not include individuals from the general population. To make this age issue clearer, we have removed the age range from the exclusion criteria and described it in the inclusion criteria. Please see lines 83-84 in the third paragraph (pages 4).

b. How about the authors clarified or measure the validity of previous or current neurological disorders, uncontrolled clinical conditions, and taking antidepressant, anxiolytic, and antipsychotic medications?

Answer Q9b R#1: We thank R#1 for this relevant question. In addition to the face-to-face questionnaire carried out by each participant, we were able to verify these exclusion criteria through data obtained in the periodic healthy examination performed every six months by the occupational physician of our Hospital. Please see lines 139-140 in the second paragraph (page 6).

Q10 R#1: Please provide more detail about the validity and reliability FACIT-Sp among Brazilian … reliability (based on your results) each domain because the author used each domain

Answer Q10 R#1. In the RV, we described the validity and reliability of FACIT-Sp among Brazilian and provided our data of the FACIT-Sp and its domains. Please see lines 107-112, first paragraph (page 5).

Q11 R#1: Line 95-97. “Normally distributed continuous variables (age and FACIT-Sp total score) are presented as the means and standard deviations and were evaluated using an independent t-tests. As well as “The associations are presented as odds ratios (OR) and 95% confidence intervals (CI 95%)”. Please provide the result, could be easier to use the table related the all objective (before, during the pandemic and each period).

Answer Q11 R#1: We fully agree with R#1. We displayed our results in a table. Then, we include “Table 1” in the RV. 

Q12 R#1: “The methods: needs to be more complete …”

Answer Q12 R#1: We thank the reviewer for pointing this out. All adjustments and additions of new sentences are in the “Material and Methods” section of the RV. Please see this section on pages 4-6 of the RV. 

Q13 R#1: “Please provide references related to the minimum sample size and the kind of assessment of sample size method.”

Answer Q13 R#1: For the regression method it is well established that 15 subjects for each predictor is desirable. Hair et al. [Hair, Black, Babin, Anderson & Tatham. Multivariate Data Analysis, 2014, 7th Edition] have suggested a minimum of 10 subjects for predictor. As four predictors were used in our model (sex, age, religious affiliation, and the total FACIT-Sp score), at least 40 subjects (n=40) were needed. For the prediction analysis of the three spirituality dimensions, at least 30 individuals (n=30) were required to investigate whether these dimensions could predict significantly AS among HCWs.

Q14 R#1: “Methods in this manuscript are generally too vague. They need to be more descriptive and clearer in defining inclusion the subjects.”

Answer Q14 R#1: In the RV, we described in detail the inclusion of the subjects. Please see lines 83-84, second paragraph (page 4), and lines 86-89, fourth paragraph (page 4).

Q15 R#1: “Line 111, please provide the detail of ethical approval ethic of this study.”

Answer Q15 R#1: We agree with R#1. In the RV, we provide more detail of the ethical approval of our study. Please see lines 133-139, second paragraph, page 6.

Q16 R#1: “In the discussion section, please describe the reason or mechanism of spirituality might protective anxiety symptoms.”

Answer Q16 R#1: In the revised version, we have included one more subsection [(“The effect of spirituality on AS starting before and during the pandemic (first and second objectives)]” to describe a potential reason or mechanism by which spirituality may protect against AS. Please see lines 220-235 (pages 10 and 11).

Q17 R#1: “Please provide data value of sex, age, and religious affiliation based on the objective”

Answer Q17 R#1: We agree entirely with R#1. We included a table (“Table 1”) in the RV that we provided our data of sex, age, and religious affiliation based on the two different starting times (AS starting before and during the pandemic). Please see these variables displayed in “Table 1”.

Q18 R#1: “Line 160. “Sex, age, and religious affiliation were not included in the logistic regression equation because we have applied the forward analysis”. Please provide more detail about the reason and previous studies to similar or contrast about the findings among HCWs (don’t general population).”

Answer Q18 R#1: We thank the reviewer for pointing this out. We agree that this sentence in line 160 of the original manuscript needs to be more descriptive. Then, we included some sentences in the RV. Previous studies during the pandemic have shown that sex and age are predictors of AS in HCWs (lines 50-51, page 3). However, it is not known that religious affiliation and spirituality are predictors of these symptoms in HCWs. Then, we used the forward stepwise regression method to find the best predictors of the AS. Our model considered sex, age, religious affiliation, and the FACIT-Sp total score to investigate whether these variables could significantly predict less AS among HCWs. Based on our approach, sex, age, and religious affiliation were not included as predictors in the logistic regression model (forward regression method). Spirituality was the only reliable predictor of AS in HCWs. Please see lines 123-126 (pages 5 and 6), and lines 202-205 (page 9). 

Q19 R#1: “Should include potential other confounding variables that could be related to spiritual and anxiety symptoms… A larger sample of healthcare workers recruited from various areas in Brazil …”.

Answer Q19 R#1: We agree with R#1. Then, we included this interesting comment in the limitations section of the RV. Please see lines 252-254 (page 11).

Reviewer #2:

“It would be best if the discussion and conclusion sections could be expanded upon to allow for a more robust discussion of results and their direct implications of future practices for the well being of HCWs. The manuscript was very robust with statistical analysis that showed the implications of spirituality on AS, however lacked a next step for HCW. This would be helpful to add to the manuscript.”

We are appreciative of the R#2 encouragement and the opportunity to revise the manuscript. Basically, R#2 raised two interesting points.

Q1 R# 2 “….best if the discussion and conclusion sections could be expanded upon to allow for a more robust discussion of results and their direct implications of future practices for the well being of HCWs.”

Answer Q1 R#2: We thank R#2 for this thoughtful comment. We expand the discussion and conclusion sections to describe a more robust discussion of the findings and their direct implications for future practice for the HCWs. Please see lines 196-200 in second paragraph (page 9); lines 222-235 (page 10 and 11); lines 242-250, second paragraph (page 11); lines 257-262, first paragraph (page 12); lines 263-266, second paragraph (page 12); lines 269-271, third paragraph (page 12); lines 275-278, fourth paragraph (page 12).

Q2R# 2 “...very robust with statistical analysis that showed the implications of spirituality on AS, however lacked a next step for HCW.”

Answer Q2 R#2: We thank R#2 for this interesting comment. Please see lines 260-262, first paragraph (page 12); lines 263-266, second paragraph (page 12); lines 277-278, fourth paragraph (page 12).

The revised manuscript incorporated all the thoughtful suggestions and comments made by the two anonymous reviewers. The manuscript is considerably improved after taking into account all the interesting points raised by the reviewers.

We hope it will now merit publication in PLOS ONE. 

Sincerely

Julio Cesar Tolentino, M.D, Ph.D.

---

## [Decision Letter · Decision Letter 1]

10 Jan 2022

PONE-D-21-18892R1Spirituality as a protective factor for anxiety symptoms in healthcare workers during the COVID-19 outbreakPLOS ONE

Dear Dr. Tolentino,

Thank you for submitting your manuscript to PLOS ONE. After careful consideration, we feel that it has merit but does not fully meet PLOS ONE’s publication criteria as it currently stands. Therefore, we invite you to submit a revised version of the manuscript that addresses the points raised during the review process.

Please address the issues concerning the rationale and methodology of the study raised by reviewer 3.

We look forward to receiving your revised manuscript.

Kind regards,

Rosemary Frey

Academic Editor

PLOS ONE

Reviewers' comments:

Reviewer's Responses to Questions

**Comments to the Author**

1. If the authors have adequately addressed your comments raised in a previous round of review and you feel that this manuscript is now acceptable for publication, you may indicate that here to bypass the “Comments to the Author” section, enter your conflict of interest statement in the “Confidential to Editor” section, and submit your "Accept" recommendation.

Reviewer #1: All comments have been addressed

Reviewer #3: All comments have been addressed

2. Is the manuscript technically sound, and do the data support the conclusions?

Reviewer #1: Yes

Reviewer #3: Partly

3. Has the statistical analysis been performed appropriately and rigorously? 

Reviewer #1: Yes

Reviewer #3: No

4. Have the authors made all data underlying the findings in their manuscript fully available?

Reviewer #1: Yes

Reviewer #3: Yes

5. Is the manuscript presented in an intelligible fashion and written in standard English?

Reviewer #1: Yes

Reviewer #3: Yes

6. Review Comments to the Author

Reviewer #1: I am pleased to read and review manuscript ID PONE-D-21-18892 entitled " Spirituality as a protective factor for anxiety symptoms in healthcare workers during the COVID-19 outbreak". The study is interesting and I suggest to publish

Reviewer #3: This manuscript describes the relevance of spirituality/religion to anxiety symptoms among Brazilian healthcare workers (HCWs) prior to and during the CV19 pandemic. The topic is novel, data (assessment of pre- and post-CV19 anxiety) is very novel, and the paper is worthwhile. However, the manuscript would benefit from a number of changes.

Abstract: Description of the study methods is unclear - the authors did NOT have two starting times rather participants were assessed at a single time-point during the CV19 pandemic and asked retrospectively about pre-CV19 symptoms of anxiety. Further, the word "converted" is inappropriate and a bit odd given that authors assessed spirituality/religion; authors should rephrase that participants were divided into those with chronic anxiety (pre-CV19 and during) vs. acute anxiety (only during CV19) and this language should be used throughout the paper. Reference to the specific dimensions of spirituality should be omitted from the abstract since readers are likely not yet familiar w/the measure or its sub-dimensions without having read the paper. Importantly: The study was conducted in Brazil yet this is omitted from the title and abstract.

Introduction: Justification of the study should be stronger - authors simply claim that age and gender are relevant and spirituality should be examined. Further, authors do not review existing research on spirituality/religion and anxiety, which are more complex than a simple buffering effect - see https://psycnet.apa.org/record/2020-20098-003 for a recent review. The rationale for looking at spirituality among HCWs is also absent. The context of the study - Brazil - should be mentioned as a justification, since spirituality/religion is commonplace within Brazilian culture. The measure need not be introduced in the introduction. Relevance of CV19 to both anxiety and spirituality should be explained: Why might spirituality be more (or less?) relevant to HCWs with anxiety that preceded CV19 vs. those experiencing acute anxiety during the pandemic without a history of anxiety? Hypotheses should be provided along these lines - e.g., Many people increase spirituality in times of distress and those without a history of anxiety may benefit less from spirituality, Alternatively, those with chronic anxiety may stand to benefit the most from spirituality. The authors need to justify their methods and approach more clearly.

Methods: Sampling method is inadequately described; was this a convenience sample? How were participants recruited? How many refused to participate? Were they compensated? More importantly, the analytic plan needs substantive revision. Why did the authors only select participants without AS before the pandemic?? As is, the authors examined the relevance of spirituality to anxiety, among participants with no significant anxiety - what is the relevance of such an approach?! Similarly they excluded individuals using psychotropic medications - why would they do this considering that the main variable under study is clinical? Another concern pertains to the measure of spirituality. As Koenig and others have explained, the FACIT assesses multiple dimensions of "spirituality" and the peace sub-scale is more akin to an assessment of mental health than spiritual/religious life. Thus, a significant negative relationship between "peace" and anxiety is not very meaningful. By contrast, the "faith" subscale would yield more interesting results. Again, I encourage the authors to examine these variables among individuals with chronic vs. acute anxiety within their sample.

Results and Discussion not reviewed in light of the above substantive concerns.

7. PLOS authors have the option to publish the peer review history of their article (what does this mean?). If published, this will include your full peer review and any attached files.

Reviewer #1: No

Reviewer #3: No

---

## [Author Response · Author response to Decision Letter 1]

24 Feb 2022

Prof. Emily Chenette, Ph.D. - Editor-in-Chief – PLOS ONE

Prof. Rosemary Frey, Ph.D. - Academic Editor – PLOS ONE

Dear Editors, 

 We are glad to know that, based on the first two anonymous reviewers, our revised manuscript entitled “Spirituality as a protective factor for chronic and acute anxiety in Brazilian healthcare workers during the COVID-19 outbreak " has merit for publication. We thank the Editors and the first two reviewers for the time spent reviewing our manuscript and for the valuable input given to improve it. The Reviewer 1 concluded that “The study is interesting and I suggest to publish” and the Reviewer 2 said that “The manuscript was very robust with statistical analysis that showed the implications of spirituality on AS”. After this first major revision, the two Reviewers agreed that all questions were correctly addressed in the revised manuscript (RM). Meanwhile, the Editors invited a third reviewer. After receiving the comments raised by all the three reviewers, the final decision was a minor review that should address the new points raised by the third Reviewer.

The Reviewer 3 (R#3) stated that “This manuscript describes the relevance of spirituality/religion to anxiety symptoms among Brazilian healthcare workers (HCWs) prior to and during the CV19 pandemic. The topic is novel, data (assessment of pre- and post-CV19 anxiety) is very novel, and the paper is worthwhile. However, the manuscript would benefit from a number of changes.”. We greatly appreciate his/her meaningful comments on our manuscript. 

Please find below the detailed point-by-point responses to all the interesting questions (Q) raised by the R#3. 

Q1: “Abstract: Description of the study methods is unclear - the authors did NOT have two starting times rather participants were assessed at a single time-point during the CV19 pandemic and asked retrospectively about pre-CV19 symptoms of anxiety. Further, the word "converted" is inappropriate and a bit odd given that authors assessed spirituality/religion; authors should rephrase that participants were divided into those with chronic anxiety (pre-CV19 and during) vs. acute anxiety (only during CV19) and this language should be used throughout the paper.”

Answer Q1: We agree with the point raised by the R#3. In the abstract of the RM, we stressed that participants were assessed at a single time-point during the pandemic. Then, we divided the subjects into chronic anxiety (anxiety that started pre-COVID-19 and persisted during the pandemic) vs. acute anxiety (anxiety that started only during the pandemic). These terms (chronic and acute anxiety) were incorporated in the abstract, figures (Fig.1 and Fig. 2), table 1, legends, title, and throughout the RM.

Q2: “Abstract: … Reference to the specific dimensions of spirituality should be omitted from the abstract since readers are likely not yet familiar w/the measure or its sub-dimensions without having read the paper.” 

Answer Q2: As our results are based on these specific dimensions, we believe that we should keep this information in the abstract.

Q3: “Importantly: The study was conducted in Brazil yet this is omitted from the title and abstract.”

Answer Q3: We entirely agree with this interesting comment raised by the R#3. Please see the revised title and abstract (line 35- page 2).

Q4: “Introduction: Justification of the study should be stronger - authors simply claim that age and gender are relevant and spirituality should be examined. Further, authors do not review existing research on spirituality/religion and anxiety, which are more complex than a simple buffering effect - see https://psycnet.apa.org/record/2020-20098-003 for a recent review. The rationale for looking at spirituality among HCWs is also absent...”. 

Answer Q4: In the revised introduction we stressed that spirituality involves cognitive and emotional states such as beliefs, motivations, and a sense of gratitude as pointed out by Rosmarin & Bethany (2020). Positive cognitive or emotional aspects of spirituality have been found to be associated with less anxiety (https://psycnet.apa.org/record/2020-20098-003). Please see lines 85-87 (page 3). Moreover, we also make clear that González-Sanguino et al. (Brain Behav Immun. 2020;87: 172–176), using regression methods, have demonstrated the importance of spirituality as the main protector against appearance of anxiety. Please see lines 88-89 (page 3).

 In the RM introduction, we fully extended the rationale for looking at spirituality among HCWs. Please see the introduction of the revised version (lines 90-93- page 3).

Q5: “Introduction … The context of the study - Brazil - should be mentioned as a justification, since spirituality/religion is commonplace within Brazilian culture.”

Answer Q5: We thank the reviewer for pointing this out. Please see lines 99-111, pages 3-4.

Q6…The measure need not be introduced in the introduction…”

Answer Q6: We still believe that a brief description of the FACIT should be present in the introduction because we extensively used the spirituality dimensions derived from this questionnaire. 

Q7: “Introduction: … Relevance of CV19 to both anxiety and spirituality should be explained: Why might spirituality be more (or less?) relevant to HCWs with anxiety that preceded CV19 vs. those experiencing acute anxiety during the pandemic without a history of anxiety? Hypotheses should be provided along these lines - e.g., Many people increase spirituality in times of distress and those without a history of anxiety may benefit less from spirituality, Alternatively, those with chronic anxiety may stand to benefit the most from spirituality. The authors need to justify their methods and approach more clearly.”

Answer Q7: In the RM we make clear that we did not perform any measure of a putative increase of spirituality. We did not assess spirituality before the pandemic. Although many people may increase spirituality in times of distress, this interesting question entails a new project. However, we pointed out that healthcare workers are under stressful conditions during the pandemic (lines 124-126, page 4).

 As there is an increase in the number of subjects with anxiety in times of distress, it is conceivable that higher basal protection should be needed to overcome the demands of a new stressful situation. Therefore, some people without anxiety symptoms preceding the COVID-19 might not be entirely protected during the pandemic. We hypothesized that subjects without pre-pandemic anxiety with higher spirituality at the baseline should be able to overcome the stressful time during the pandemic. Therefore, given the potential protection of greater spirituality against anxiety [12–16], we hypothesized that those experiencing acute anxiety during the pandemic would exhibit lower basal spirituality, as compared to those who remained without anxiety before and during the pandemic. 

 In addition, differences in spirituality scores would reflect differences in specific spirituality dimensions. As personal faith can increase psychological resilience [45,46], we hypothesized that faith would reach the highest score in those without chronic and acute anxiety. In addition, as faith is associated with better coping in stressful times [47–49], we also hypothesized that higher personal faith could be a potential predictor of less acute anxiety during the pandemic. Please see lines 131-134, second paragraph (page 4).

 We would like to stress that we have used the definitions according to the R#3 suggestions (“…authors should rephrase that participants were divided into those with chronic anxiety (pre-CV19 and during) vs. acute anxiety (only during CV19) and this language should be used throughout the paper.”)

Q8: “Methods: Sampling method is inadequately described; was this a convenience sample? How were participants recruited? How many refused to participate? Were they compensated?”

Answer Q8: In the Methods section of the RM, we answered these interesting questions raised by the R#3. Please see lines 157-158, third paragraph (page 5), lines 227-228, second paragraph (page 8), and lines 238-239, third paragraph (page 8).

Q9: “… Why did the authors only select participants without AS before the pandemic??” 

As is, the authors examined the relevance of spirituality to anxiety, among participants with no significant anxiety - what is the relevance of such an approach?!”

Answer Q9: Thanks for your comment, but we aimed to investigate whether spirituality would predict anxiety that started before the pandemic and persisted during the pandemic (chronic anxiety) and anxiety that started only during the pandemic (acute anxiety). 

 We select participants without anxiety symptoms before the pandemic in order to investigate the predictors of acute anxiety during the pandemic- Please see lines 172-176, first paragraph (page 6). 

 Again, we did not measure spirituality before the pandemic because it was not included in our objectives. 

Q10: “Similarly they excluded individuals using psychotropic medications - why would they do this considering that the main variable under study is clinical?”

Answer Q10: We excluded individuals using psychotropic medications to avoid a potential protective effect of these drugs on the development of anxiety symptoms, especially in those who started these symptoms only during the pandemic (acute anxiety). Furthermore, it should be mentioned that these medications could also be used to treat some clinical diseases.

Q11: “Another concern pertains to the measure of spirituality. As Koenig and others have explained, the FACIT assesses multiple dimensions of "spirituality" and the peace sub-scale is more akin to an assessment of mental health than spiritual/religious life. Thus, a significant negative relationship between "peace" and anxiety is not very meaningful. By contrast, the "faith" subscale would yield more interesting results. Again, I encourage the authors to examine these variables among individuals with chronic vs. acute anxiety within their sample.”

Answer Q11: We thank the R#3 for pointing this out. Previous studies on the peace domain derived from the FACIT-SP have been shown that this dimension reflects an affective component of spirituality [35,36,41,42]. In addition, the peace dimension correlates with mental health [11,43,44], and some researchers have reported a significant negative relationship between "peace" and anxiety [11,43]. Furthermore, we agree with the R#3 that the findings of the present study about the faith spirituality dimension are more interesting. Indeed, we found that the faith dimension emerged together with peace to confer protection on acute anxiety starting only during the COVID-19 outbreak. 

 As suggest by R#3 we examined these variables (peace and faith spirituality dimensions) in the fourth objective of this study. 

Q12: “Results and Discussion ….”

Answer Q12: In the RM we made the changes based on the interesting comments raised by the R#3. In addition, we have included one more subsection in the Results and Discussion sections regarding the fourth objective. Please see the Results and Discussion sections in the revised version. 

The revised manuscript incorporated all the thoughtful suggestions and comments made by the three anonymous reviewers. After responding to the new points raised by the R#3, we hope it will now merit publication in PLOS ONE.

Sincerely

Julio Cesar Tolentino, M.D, Ph.D.

---

## [Decision Letter · Decision Letter 2]

12 Apr 2022

Spirituality as a protective factor for chronic and acute anxiety in Brazilian healthcare workers during the COVID-19 outbreak

PONE-D-21-18892R2

Dear Dr. Tolentino,

We’re pleased to inform you that your manuscript has been judged scientifically suitable for publication and will be formally accepted for publication once it meets all outstanding technical requirements.

Kind regards,

Rosemary Frey

Academic Editor

PLOS ONE

Additional Editor Comments (optional):

Reviewers' comments:

Reviewer's Responses to Questions

**Comments to the Author**

1. If the authors have adequately addressed your comments raised in a previous round of review and you feel that this manuscript is now acceptable for publication, you may indicate that here to bypass the “Comments to the Author” section, enter your conflict of interest statement in the “Confidential to Editor” section, and submit your "Accept" recommendation.

Reviewer #1: All comments have been addressed

Reviewer #4: All comments have been addressed

2. Is the manuscript technically sound, and do the data support the conclusions?

Reviewer #1: Yes

Reviewer #4: Yes

3. Has the statistical analysis been performed appropriately and rigorously? 

Reviewer #1: Yes

Reviewer #4: Yes

4. Have the authors made all data underlying the findings in their manuscript fully available?

Reviewer #1: Yes

Reviewer #4: Yes

5. Is the manuscript presented in an intelligible fashion and written in standard English?

Reviewer #1: Yes

Reviewer #4: Yes

6. Review Comments to the Author

Reviewer #1: I am pleased to read and review manuscript ID PONE-D-21-18892 revision-2. The study is interesting and I suggest to publish

Reviewer #4: researchers have corrected and answered clearly all the comments of previous reviewers. there are only two questions that must be answered clarified by the researcher

Q1; exclusion criteria; what is the reason you use the exclusion criteria regarding regularly performed every six months? describe in more detail. Line 175 Page 8

Q2; What kind of psychosocial intervention is more suitable for spiritual uplifting? provide some more specific recommendations. Line 336 page 15

7. PLOS authors have the option to publish the peer review history of their article (what does this mean?). If published, this will include your full peer review and any attached files.

Reviewer #1: No

Reviewer #4: No

---

## [Editor Report · Acceptance letter]

21 Apr 2022

PONE-D-21-18892R2 

Spirituality as a protective factor for chronic and acute anxiety in Brazilian healthcare workers during the COVID-19 outbreak 

Dear Dr. Tolentino:

I'm pleased to inform you that your manuscript has been deemed suitable for publication in PLOS ONE. Congratulations! Your manuscript is now with our production department. 

Kind regards, 

on behalf of

Dr. Rosemary Frey 

Academic Editor

PLOS ONE